# Resilience and Anti-Stress during COVID-19 Isolation in Spain: An Analysis through Audiovisual Spots

**DOI:** 10.3390/ijerph17238876

**Published:** 2020-11-29

**Authors:** Fernando Olivares-Delgado, Patricia P. Iglesias-Sánchez, María Teresa Benlloch-Osuna, Carlos de las Heras-Pedrosa, Carmen Jambrino-Maldonado

**Affiliations:** 1Department of Communication and Social Psychology, Faculty of Economics and Business Studies, Universidad de Alicante, 03080 Alicante, Spain; f.olivares@ua.es; 2Department of Business and Administration, Faculty of Economics and Business Studies, Universidad de Málaga, 29071 Málaga, Spain; mcjambrino@uma.es; 3Department of Communication Sciences, Faculty of Human and Social Sciences, Universitat Jaume I., 12071 Castellón de la Plana, Spain; benlloch@uji.es; 4Department of Audiovisual Communication and Advertising, Faculty of Communications Sciences, Universidad de Málaga, 29071 Málaga, Spain; cheras@uma.es

**Keywords:** COVID-19, coronavirus, isolation, confinement, resilience, stress, corporate ads, health communication, risk communication, public health

## Abstract

This study analyses broadcasted advertising spots during the COVID-19 isolation period in Spain. It aims to identify the narrative communicative resources and messages spread by companies/brands under the background of a global pandemic, where a common pattern highlighting the social function of brands is supposed, specifically regarding stress and resilience. We propose a mixed method based on the combination of qualitative analysis of the content of a compendium of 71 commercials and statistical analysis to group and test the correlations between some key variables, thus incorporating multivariate analysis with a quantitative method. Our main finding is the collective trend and communicative behaviour in the advertising of companies/brands during isolation, in which a change in the traditional role of advertising communication has occurred, where advertisers have become a key support in combatting the disease and a key support for health and psychological management in the Spanish population. In fact, they act as guardians of resilience and promoters for alleviating stress.

## 1. Introduction

COVID-19 has led to a global pandemic with catastrophic consequences for society [1], collapsing health systems in different countries [2] and generating a severe economic recession worldwide [3,4].

Globalization and the international movement of people have been challenged by this unprecedented health crisis, due to its rapid expansion around the world. As a result, governments such as those of China, Italy, and Spain [5] have taken drastic measures to close regions or borders and have confined their populations, in order to guarantee the health of their citizens.

Spain is one of the countries most affected by the pandemic in Europe. For this reason, the Spanish government decreed a state of alert at the national level and imposed unprecedented measures on the country. This led to the isolation of its population in their homes from 14 March to 4 May 2020. Subsequently, a phased de-escalation began, which ended in all regions on 21 June 2020.

The coronavirus has transformed all areas of life in just a few months: personal relationships, family, health, the economy, and work. Moreover, never before has there been such a systemic and global situation, in which public health and economic crises are combined with social hibernation, as in the obligatory confinement in Spain in the spring of 2020. This shock has not been easy to assimilate in a balanced way by all human beings: in the face of this uncertain situation and 51 days of isolation, traumas and mental health problems have arisen [6] and the phenomenon of a social contagion has even been observed [7].

In addition, Spain has produced the highest death and injury rates in the world with far-reaching socio-economic and employment impacts, according to Data Insights Solutions 40 dB [8].

Another added risk factor is the information or misinformation that the population is subject to. This circumstance is in contrast with the typical dependence on communication in health crisis situations [9,10], as well as the expansion of the use and consumption of information [11,12]. For these reasons, communication and messages have created negative consequences for many populations [13] and, somehow, have become a source of stress for society in this unprecedented situation. Therefore, communication plays a prominent role, as the central nervous system of our society [14]. The World Health Organization [15] has highlighted communication as one of the greatest challenges and identified risk communication as one of the core competencies required to cope with a pandemic. The ability to transmit information quickly and clearly on different media platforms is essential for the effective management of a public health emergency.

This work is framed within the field of risk communication, particularly that relating to public health, for which the literature has a certain tradition of health communication and media studies [16,17].

According to the previous literature on risk communication, a person in a health crisis confronts the known/unknown, which causes them to remain hesitant and to have to manage uncertainty regarding each message and its specific context [18,19]. Furthermore, people have very diverse capacities to process risk messages, including scientific and technical understandings of risk. Different levels of tolerance or risk perception among different people are due to ambiguity in public health issues, multiple sources of information, fake news transmitted through social networks, and so on [12].

The literature has also established that communication is paramount in risk management [20,21,22]. Authorities should establish common communication channels and languages with the population for effective risk communication [23,24]. Experts use technical or scientific terms, while citizens perceive risks more emotionally [4]. Perception is crucial, as risks become real once they are perceived as such [25].

There is extensive literature on health crisis management, focused on communication [17,22,26]. However, the COVID-19 global pandemic is unprecedented in the recent history of the country and, so, the present research should provide valuable information regarding the ability of Spanish organizations to communicate effectively with their audiences and show their support to society for the solution of economic and social problems derived from this health crisis. There has been hardly any research on the social function that brands can play in public health and welfare through their corporate advertising campaigns; hence, leading to the relevance of our contribution. It is precisely the possibility of checking whether, in a crisis of health and social alert, brands act and develop a favourable narrative in their communications, in accordance with the needs and means of understanding/facing this situation, that motivated the development of this research.

### 1.1. Stress during Isolation by COVID-19

A pandemic is a potential source of stress for people [27]. This pandemic has led to increased levels of uncertainty, fear, anxiety, distress, and mental health problems [4]. In addition, social confinement has increased the reasons for stress, as there exists an internal struggle of feelings and moods associated with the development of the pandemic itself, as well as with measures to contain it [7]. Brooks et al. [6] stated that quarantine and confinement can affect the physical and mental health of people who suffer from it, as it is a disturbing and unbalanced experience with negative psychological effects. Isolation, even home confinement, has an impact on our balance (especially with respect to stress) and, in order to reduce possible negative effects, the authorities must correctly explain the reasons for isolation, make it as tolerable as possible, and adhere to the announced dates; otherwise, the general mood may be altered. A perception of improvisation increases uncertainty [28]. Coping strategies can offer a proactive response to combat the threat of such stressors [29] (see Figure 1). On the other hand, communication and continuous contact (especially through social networks between citizens) can favour both the transmission of feelings and attitudes and the development of collective therapies to overcome them [7].

### 1.2. Resilience during COVID-19 Isolation

Exposure to disasters has been associated with a variety of mental health consequences [30]. Resilience is a concept studied in the social sciences, which has been emerging in the disaster mental health literature [31,32,33], as it allows us to seek opportunities in turbulent environments and to emerge victorious and strengthened from any type of crisis. Developing the capacity to react and function after a traumatic event is common and characteristic of normal coping and adaptation [31].

Resilience entails not only the ability of the human being to face and overcome life’s adversities, but also to be positively transformed by them [27,34,35]. It implies personal growth [36] and/or allows one to extract a positive and enriching reading after a negative experience. According to Grotberg [37], the sources of resilience are related to four aspects: (1) having help when you are in danger, (2) being happy when you do something positive for others, (3) being sure that everything will be okay, and (4) being able to talk to someone and communicate to find their support. Walsh [38] proposed the concept of family resilience, which refers to the intrafamilial process that sustains the resilience of all its members. Finally, Adger ([39], p.347) emphasized the concept of social resilience as “the ability of groups or communities to cope with external stresses and disturbances as a result of social, political and environmental change”. This type of resilience has its maximum expression when the functioning of a system, environment, or social structure and its stability, as understood and operated, undergoes a substantial change [40,41,42,43]. In this case, collective social support comes into play, one of the factors upon which this research is based; see Figure 2.

The pillars or sources of resilience are experiences that allow individuals to emerge successfully from an adverse or traumatic situation. One concept of interest, also defined by Cyrulnik [36], is “tutors of resilience”, referring to those who support others; those who need it most. Caregivers have the task of generating hope and nurturing relationships and emotional security [44]. In the case of social resilience, institutions recognized by society, such as governments, authorities, the media, or businesses, can better cope with crisis situations [45]. Communication is a central component of most, if not all, community resilience models [46]. Norris et al. [33] included communication and information as some of the core adaptive capacities that constitute community resilience. Such adaptive capacity includes the attributes of narratives, responsible media, skills and infrastructure, and trusted sources of information.

### 1.3. Media Consumption and Advertising during Confinement by COVID-19 in Spain

According to Xifra [47], health crises require reliable spokespeople. This aspect is essential in both public and governmental communications, as seen in the first weeks of the pandemic, and is key to understanding that the traditional media, in these global crisis situations, act as a refuge and generate more confidence, as their tradition is an attribute that strengthens their relationship with the public [48].

Edelman [49] conducted research in 12 countries during the second week of March 2020 regarding COVID-19, confirming the crucial role played by traditional mass media during crises. In Spain, the months of March, April, and May had historical records in media consumption, especially for television. In addition to television coverage, 33.6 million Spaniards consumed this medium daily, representing 74.2% of the population [4].

Generalist television consumption in Spain increased by 51 min a day during the state of alarm and confinement, with respect to data from the same period (14 March to 20 June) of the previous year (Figure 3).

The first weekend of isolation in Spain increased television consumption, breaking historical records at 413 min per person, 67% more than the previous weekend [51]. This trend continued during March, April, and May, with record television consumption in Spain; increasing by 89 min in March, 69 min in April, and 37 min in May, with respect to the same months in 2019 (Figure 3).

Due to the uncertainty regarding the situation and home isolation, Spanish citizens consumed more television. Thus, the months of March and April became those with the highest television audience in Spanish history. Data from March showed an average consumption of 282 min per person per day (4 h and 42 min). The average length for people who had watched TV for at least one minute a day was 369 min (6 h and 9 min) [52]. The progression in the television audience continued in the month of April, with numbers never seen before in the conventional Spanish television, with 302 min (5 h and 2 min) and 395 min (6 h and 35 min), respectively [4,53]. In the month of May, the consumption of television—although low with respect to the previous months—remained high at 250 min (4 h and 20 min) [54].

It can be concluded that free-to-air or generalist television was the most consumed medium in Spanish society during isolation. More than half of Spaniards (54.6%) chose television as their first channel of communication when looking for truthfulness and proven credibility, followed by the press [55]. This is why the role of broadcasted advertising on television played an important role at this time.

Paradoxically, the number of advertising campaigns decreased during confinement and it was not until the beginning of June that they increased by 15%, due to the arrival of advertising from sectors that had suspended their campaigns during confinement, such as the automotive sector and breweries, which then intensified their presence in the summer.

During a health crisis such as the one we are currently dealing with, the establishment of trusting relationships with the public becomes more relevant. Xifra [47] indicated that this is not the time to sell, but rather to inform and offer solutions, such that brands should focus their efforts on finding appropriate and meaningful solutions to the problems faced by those directly or indirectly affected by COVID-19. 

Table 1 shows the opinion of the Spanish population regarding the role that brands must play in times of crisis. It can be seen that the majority of Spaniards agreed that branded advertisements should support society in these difficult times. Thus, 83% of the population considered that advertising should show how companies can be useful in the new daily life, while 79% considered that advertising should use a reassuring tone supporting resilience and anti-stress messages, in response to confinement.

Based on the above, the purpose of this research was to discover whether audiovisual business communication broadcasts through mass channels (e.g., television) have played a social function during isolation, particularly regarding their service to public health, in this exceptional and adverse situation caused by the COVID-19 pandemic.

The following research questions were raised:-RQ1. Have the brands changed their function for supporting the dynamics of the consumer society to a social function of support in containing COVID-19 and managing the state of mental and psychological health of their audiences during confinement?-RQ2. Have companies and their brands incorporated, through the audiovisual narrative of their advertising, positive messages in favour of individual and collective resilience in Spain?-RQ3. Have companies and their brands incorporated messages that could serve as coping strategies against the psychosocial stress produced by the pandemic?

## 2. Materials and Methods

### 2.1. Measures and Instruments

A mixed method was carried out for the purpose of content analysis of (audiovisual format) advertising broadcasted during the isolation period in Spain. This first choice was a qualitative method while, from the qualitative point of view, a statistical analysis with quantitative information was applied, the highlights of which are correlation analysis and grouping by common patterns of communicative messages. This combination is novel, with respect to the existing literature, and allows for obtaining a holistic vision of the phenomena.

The content analysis of audiovisual spots makes it possible to identify resilience and mechanisms to react/alleviate the stress derived from the effect of the pandemic, as well as from isolation. The videos were compiled and coded using a quantitative approach, through content analysis [58]. The messages, images, and even music used in the advertisements served to evaluate the social function of companies and their brands during the outbreak. We sought to determine whether the discourse and other resources contributed to Spanish society, in terms of mental health management for the wider population, focusing mainly on resilience and stress. Furthermore, the comparison of advertisements allowed us to study the existence of social resilience sentiments [39] and other sentiments in the first phase of the COVID-19 crisis [4,7], through the observable features of the written texts, images, and videos embedded in the messages.

Table 2 shows the evaluation sheet for content analysis based on the works of several authors, highlighting (for measuring resilience) Connor and Davidson [35], Walsh [38], Cyrulnik [36], Grotberg [37], and (for stress) Brooks [6]. Additionally, some adaptations were introduced, in order to adjust the measurement instrument to the unprecedented situation derived from this global pandemic. In brief, the analysis content sheet took into account the following variables: “date of issue”, “duration of the spot”, “medium or platform”, “brand name”, “brand name”, “type of message”, “Sector of activity”, “slogan”, “verbal content”, “stress”, “resilience”, “tone of the message”, “voice over”, and “emotional burden” (Table 2).

There were three types of variables: Ordinal variables, such as the length of the advertising; Likert scale variables with five points, for tone and emotional burden; and, finally, nominal variables for resilience and stress.

Furthermore, a correlation analysis, analysis of variance (ANOVA), and conglomerates of spots by frequency were carried out, in order to test the relationships between different variables. Despite the fact that qualitative analysis was the core of this study, quantitative analysis offers a complete vision and can explain the relationships and coincidences of strategies in the audiovisual advertising generated during isolation period.

The coding was carried out by six expert researchers, in order to avoid intersubjectivity. The reliability was ensured by discussing the data in the cases of discordance between them and crossing random entry of data accordingly [59].

### 2.2. Sample

A detailed observation of 71 advertising spots with their 1350 shots served as the basis for the analysis. They were all the audiovisual brand advertisements that were aired on TV, IGTV, and/or social media during the isolation in Spain. Consequently, the sample consisted entirely of commercial communication developed in an unprecedented time for both society and brands. It should be pointed out that institutional advertisements from the Government, particularly from the Health Ministry and Non-Governmental Organizations, were not considered, due to the goals of this study. By contrast, the promotion of tourist destinations was included, due their notable presence in advertising. Therefore, the focus of advertising by enterprises in an unprecedented situation of outbreak was analysed.

The analysis focused on advertising spots corresponding to TV consumption during the isolation period in Spain. Furthermore, IGTV and social media were simultaneously used as platforms to broadcast advertisements; although most of these campaigns had versions airing on other mass media platforms.

A total of 71 advertising spots were considered in this case. It was necessary to choose a random sample, as it was possible with the total volume of broadcasted commercials. Nevertheless, they were grouped by sector of activities, in order to ensure that all representative economic sectors were taken into account (see Figure 4). It has been stated that the analysis of several cases in the same study might yield more accurate results [60]. Therefore, analysis of the same issue in many companies/brands not only allows for a comparison that adds value to our research, but also tracks the practical implications for the design of communication strategies in crises. To date, the investigations with this focus remains scarce.

### 2.3. Data Collection

Data were collected over 8 weeks (51 days) of the outbreak, specifically during the confinement period in Spain from the 14th of March (date of isolation start point in Spain) to the 4th of May (date when the health authorities heralded a new phase in the pandemic).

### 2.4. Validity and Reliability

The validity and reliability of this study were supported mainly in the three following ways:The content analysis was carried out with all the advertising spots broadcasted by companies in the targeted period;There was a representative sample of commercials from different sectors. Therefore, the study does not impose any sample frame bias, regarding a certain advertising style or kind of resource;The application of statistical analysis added value and increased the robustness of the results.

## 3. Results

The isolation period spurred a common pattern of advertising communication by firms. On one hand, the content focused on encouragement and solidarity messages. On the other hand, the most used communicative line was that of searching for early recovery. Both showed empathy to the sentiments and problems of the population in this unprecedented outbreak. Consequently, companies took an active role to help overcome the situation linked with the pandemic. In fact, the most creative axis in advertising spots offered encouragement through reassuring or supporting messages for facing the situation experienced as a consequence of isolation (68%) and less regarding the devastating consequences of the disease (i.e., COVID-19) directly. Likewise, 32% of messages were designed in order to create and strengthen public awareness and social sensitivity on the prevention of the contagion. In line with the traditional tone of advertising campaigns, even in a crisis of such an historic scale, the messages mostly had a very positive (54%) or positive (15%) tone, as compared with the scarce representation of negative (3%) and very negative (1%) messages. The rest of the communications adopted a realistic tone (27%). However, by contrast with the ‘normal’ typology of advertising, in commercial messages during the isolation period in Spain, companies/brands resorted to corporate (71%) or hybrid (29%) advertising, but there was no representation in the sample of spots of messages entirely oriented towards sales. Furthermore, there were no differences in any economic activity. Figure 5 shows the typology of messages, disaggregated by sectors.

Based on our findings, the burden charge was high. The values were concentrated at the top of the scale—that is, very emotional (25%) or emotional (35%)—while only 15% were little or rather emotional; there was a single ad that was coded as <not at all emotional> (Figure 6). The messages, the images, and even the music and voiceovers were chosen to strike at the heart of the targeted people.

Regarding the content of the messages embedded in advertising spots, the coincidences in the topics should be emphasized. Word clouds are very illustrative, in this sense. Figure 7 is the resulting word cloud of the slogans used and Figure 8 shows the word cloud of the whole text (listened and written) of the spots.

The most cited words were related to “we” and “you” (second person plural), evoking the collective and personally targeting the public in the message. In the same line, the images accompanying these advertising spots tried to reflect the reality of the public: family, friends, and the closest to their social circles. The brands even made an effort to constitute in their messages the “we, our, us” of the public. In any way, they wished to show that they were part of the situation and that they suffered and empathized with the lived experience of viewers. The second group of most-used words comprised: “together”, coherently with previous word; “home” and “stay at home”. The latter term coincided with one of the most global hashtags during the pandemic in Spain and in all affected countries. Additionally, ideas linked to overcoming the pandemic—literally, “to overcome” and “will win”, “to make it possible”, “it will end”, “to resist” and “to can”—had outstanding presence in the slogans. All of these are directly concerned with resilience. Finally, the words “soon” and “thank you” were significant in the slogans used. It should be emphasized that the words that directly addressed the disease never appear in the analysed ads (i.e., “COVID-19”, “coronavirus”, “disease”, or “pandemic”). Even the word “health” appeared only once in all slogans during the isolation.

There was some coherence between the most referred-to concepts in slogans and those in the entire messages of the advertising (written or spoken). In the same line, positive encouragement words prevailed and, consequently, the idea of overcoming the health crisis and returning to normality—especially regarding social life—were notably used. Likewise, the sense of “group” was emphasized. Thus, words such as “together”, “we”, and “you” were, again, often used. Furthermore, the personal call, using the first person of plural and second of plural, as well as (but less) the first person singular, was emphasized. In general, the more significant formal verb was the future “will”. It was observed that this indicates an ability to achieve common goals, overcome the crisis, and have a life similar to that which people missed. “Will” has more positive charge than the past form “did”, although it can be used to express the same idea. The words with higher frequency in the messages (excluding personal pronouns) were, in decreasing order: “home”, “now”, ”again”, “can”, “Spain”, “stay at home”, “bar”, “together”, “close”, and “people”. Regarding this list, three deserve to be highlighted: the adverbs “now” and “again”, showing that the moment to fight against the virus was precisely that, while “again” was linked to the idea of coming back to enjoy certain things that were impossible due the measurements of containment. The word “close” referred to, in most cases, social distance, but in a positive sense; for example: “be further to be close”. Following on from that, it should be noted that words with negative charge always appeared to reinforce positive messages; for example: “never loss”, “keep the distance to be more together”, and “have to meet less to overcome it”. Additionally, the word “bar” had a significant presence, showing empathy in accordance with the especially dramatic effects in the catering industry. Finally, despite relatively less representation over the sample, words such as “stop”, “live/life”, “health”, and “kisses and/or hugs” should be stressed as communicative resources to engage with the public. As general considerations, it should be stressed the presence of epic verbs such as “resist”, “fight”, “combat”, and “win”. The nostalgia for missed things is an outstanding resource in advertising during isolation. Furthermore, messages were linked with the enormous will to come back to life as usual, with verbs of comeback predominating. Finally, the immediacy linked with consumption was used in the isolation period, reflecting the social anxiety and the consequent desire for isolation and the whole pandemic to be finished.

It is at this point that the test of reliability of the scale variables was carried out and correlation analysis between pairs of variables was applied. The Cronbach’s alpha resulted in the minimum value to accept it (0.69). By contrast, the correlations were significant for tone and de-stress and between de-stress and resilience at the same level (0.005), but were not valid for burden charge with any of the scale variables chosen. In any event, the direct relationship between de-stressing and resilience levels provided evidence for the connection between these psychological issues in practice and in communication resources as well.

In view of the main issues regarding psychological health in COVID-19 times (i.e., stress and resilience), the advertising messages aimed to reduce and manage stress during the isolation and to promote resilience are pointed out in Table 3 and Table 4, respectively. On one hand, based on the stressing factors proposed by Brooks et al. [6], communicative resources were proposed to overcome those fears and mental stressors. On the other hand, the sources of resilience of Walsh [38], Cyrulnik [36], and Grotberg [37] were illustrated through the advertising narrative and, consequently, these analyses were useful in responding to research questions 2 and 3 of this study.

The application of frequency analysis to the communicative resources used in advertising spots provided different groups of narratives for de-stressing and resilience in their messages and images. First, focusing on the factors of stress the strategies/actions to face it conveyed through audiovisual narratives were grouped into three main clusters. As can be seen in Figure 9, each cluster has a long distance regarding the others and any sharing the main resources and topic formed a creative axis. The first group clustered ads that evoked the sense of social cohesion, a sense of belonging, and the acknowledgement and the appreciation of the work of staff on the front lines of the pandemic, highlighting health workers in particular. The second group was called <today and expected leisure spare and new labour reality>. These conglomerates aimed to communicate how to foster fun time: playing music, singing, dancing, cooking, parties, and celebrations. Additionally, on one hand, the plans to do after isolation had repeated motifs (e.g., travelling, walking, and going out with friends). On the other hand, there were proposals to do things differently when the health crisis has finalized, as a consequence of the awareness of those things that people missed which, before COVID-19, were normal. Even the development of a feeling of solidarity during isolation and maintaining it when the alert state would end. Likewise, scenes that showed the new lifestyle of working at home were quite common in the narratives developed by advertisers. <Routines and common lifestyle> is the name of the last group. The set of ads in this group showed everyday scenes; that is, normal routines but under the effect of isolation. Additionally, a proposal to change this routine with challenges for daily tasks (e.g., making Tik-Tok videos) should be precisely stressed. 

All communicative resources provided means to face and overcome the five stress factors identified by Brooks et al. [6] associated with the drastic measures of containment imposed under the state of emergency. Regarding resilience, Figure 10 shows fewer differences between the extremes of the two groups of sources of resilience in the advertising spots. The first group built their narrative on confidence (self-confidence, confidence in the crisis being under control and will end soon) and the need to be encouraged and resist. The second group provided a great deal of communicative resources to promote resilience: the support of social and personal relationships, knowledge/information, optimism, self-control, and the need for empathy; for example, saying thanks, taking care of the most vulnerable people, and to be positive/optimistic.

These results provide evidence regarding how companies/brands can spread messages that help the population, in terms of both of the key psychological issues of resilience and de-stress, and the importance both acquire, in terms of the creative axis, in advertising campaigns broadcasted during the phase of isolation.

## 4. Discussion

The social functions provided by company/brand advertisements for facing the psychological impacts of isolation and support for containment of a pandemic disease were highlighted in this study. On one hand, the previous mental health literature has widely dealt with their direct impact on the sentiments and mental health of people [27]. On the other hand, studies of management crisis communication, and particularly health crisis communication, have highlighted the challenges faced by governmental and public authorities in such situations [17,21]. However, the participation of companies—through their advertising—in better facing and overcoming crises from a psychological point of view has not yet been explicitly addressed. In any event, media information [20] and social media communication [4,7,10,12] have been occasionally taken into account from this point of view. Thus, this insight provides a new approach to the analysis of crises and their communication and contributes to determining the supporting role of brands in situations such the present global pandemic. The communicative resources and creative axis used during isolation by brands in Spain coincided with the integrative model of communication in crisis proposed by Reynolds and Seeger [21]. It is clear, from the outcome of this study, that brands changed their line of communication during isolation and have left their commercial goals in favour of arguments, icons, and stories directly linked with the health crisis. Consequently, research question 1 was answered, confirming the radical change in function of advertising. The sample of spots was focused on the design of communicative strategies relating to facing the uncertainty and psychological pressures due to the strict measures of isolation to stop the virus contagion in Spain. In fact, many of the narratives used were focused on managing stress and promoting social resilience; these two issues are related to research questions 2 and 3, as made evident by Table 3 and Table 4, respectively.

Companies and brands have played a social role, acting in the public interest against the pandemic and having a direct effect on the health and psychological comfort of society. All the spots broadcasted during isolation shared a common pattern, in this sense, and tried to act as resilience guardians, following the concept introduced by Cyrulnik [36]. Thus, we found other agents (brands) who can develop this activity. These companies voluntarily waived their commercial purposes and prioritized social ones. They built messages geared towards the particular circumstances and, as such, informative, educational, and awareness campaigns became prevalent in the months of the isolation. This choice was in line with governmental and health institutions during crises, as evidenced by the previous literature [20,21,22,61]. Therefore, this study contributes to showing that the models of communication in crisis from institutional and business approaches are quite similar. In any event, it seems that advertising companies focused more on psychosocial and mental health than strictly general health. Mental and physical health in challenging times, such as isolation, should be managed, as Brooks et al. [6] has emphasized. This was evidenced by the results of this work, mediated through companies and brands. In fact, the recent literature has analysed contagion sentiments in general during the pandemic [4,10] and how social media can act as collective therapy for emotional management [7]. By contrast, this study paid attention to two specific psychological issues—resilience and stress—which is our main contribution to the literature.

Precisely, these insights allowed us to answer research questions related to resilience and stress. The creative axis struggled in encouraging and inspiring coping strategies against this traumatic background, due to COVID-19. In fact, the analysed advertisements during isolation promoted messages with great charge of resilience and focused on alleviating stress. The advertising broadcasted during the isolation in Spain demonstrated that advertisements focused on the resilience sources studied by Connor and Davidson [35] and Grotberg [34,37], with different kind of resilience even being promoted in this specific dramatic crisis, following and in line with the literature: individual resilience [36], familiar resilience [38] and, finally, social-collective resilience [39].

Hereafter, we provide examples of the kind of messages used for searching to encourage discussion regarding the previous literature:-Dissuasive messages, such as “stay at home”, focused on the key measures for contention of the disease. This message was built in the line of promoting protection of the self and the collective under the environment of pandemic stress, as seen in studies such as that of Roberts and Veil [17].-Messages to overcome fears and to foster hopes, such as “everything will be fine”, increasing the feeling of security and protection. This issue has been pointed out by Reynolds and Seeger [21], regarding crisis communication management; by Grotberg [34,37], specifically focused on resilience; and by Brooks et al. [6], as linked with stress.-Ideas to combat frustration and boredom, promoting physical activity such as gymnastics, fitness, or dance. This insight has commonalities with the conclusions reached by Frigon [32]. These proposals are especially useful to manage the levels of stress associated with isolation.-Proposals for creative, playful, humorous, or challenging activities (e.g., uploading to Tik-Tok) to do at home. Strategies for finding new options, activities, and even routines in traumatic and stressing situations have been stressed as a source of resilience by Connor and Davidson [35], as well as by Goldman and Galea [27].-Messages to reinforce the idea of a nuclear family, by fostering family relationships in a group and more contact with loved ones, such as children and especially grandparents, in line with Walsh [38]. Moreover, they are coherent with the resources to reduce stress proposed by Brooks [6].-Messages fostering the imagination of future moments of happiness or the memory of what made the Spanish population feel good. Imagination is decisive in escaping from traumatic situations or experiences. This coincides with the resources to promote resilience proposed by Grotberg [34,37].-Messages whose core is social union, help, and altruism, according to Adger [39].

Therefore, companies—in line with institutions with authority in health care issues—have contributed to the prevention and contention of the disease; further, their role regarding psychosocial health during the pandemic should be specifically highlighted. This is precisely the main difference from the field of health crisis communication management, as the focus on the agent involves companies and brands, not institutions.

### 4.1. Limitations and Future Research Directions

Despite the analysis of a complete repository of ads broadcasted during isolation, the use of a larger sample would be recommended, through introducing an international comparison or even an analysis of the different narratives presented in the different waves of confinement measures during the pandemic, as well as in the future period of the “new normal”. Likewise, it would be interesting to analyse the same sample under a different approach: for example, economic, cultural, and/or psychosocial (e.g., lifestyle, family model, gender). Finally, the introduction of an audience, with three purposes: (1) to test whether exposure to audiovisual advertising has had an effect on their resilience and stress management, which should be a challenging purpose for future research works; (2) to prove if the reputation and purchasing decisions of brands have been positively influenced as a consequence of the change in the orientation of the campaign, with a clearer social component during isolation; and (3) to compare the credibility, confidence, and understanding of the communication relating to COVID-19 broadcasted by public institutions and health authorities and that carried out by companies.

### 4.2. Practical Implications

This study provides a new framework to explore how to integrate companies, through their advertising, into the health crisis communication field, engaging and reacting to the whole population. The social functions developed by companies and brands during this pandemic, specifically during isolation, regarding psychological and mental health, seem sufficient to suggest a joint effort between the government and companies. Therefore, institutional and health authorities could take advantage of the awareness of companies during crises such as the COVID-19 pandemic, in order to successfully achieve prevention and contention of the disease, along with the psychological comfort of the citizens. Moreover, brands can seize their empathy and involvement with the outbreak to gain position and reputation in the minds of consumers. Consequently, the lesson to learn from this health crisis for firms is that it poses an extraordinary challenge, from a commercial communications perspective.

## 5. Conclusions

The present pandemic has brought a significant amount of change to the social function of commercial communications during the isolation period in Spain. The dramatic global effects of COVID-19 have led to promoted advertising spots opting for emotional campaigns focusing on the lived experience and the hope of breaking the health crisis. Therefore, brands have sought to play an active role, spreading messages of encouragement and even including resources contributing to public measures for the contention and prevention of the disease in their narrative, as suggested in our research question 1. Generally, companies have set aside their commercial goals in favour of overcoming the outbreak. In fact, we found a repository of ads with emphasis on supporting the greater society in management of their psychological comfort, regarding stress and resilience as a consequence of confinement measures. Thus, brands have promoted, through their creative axis, strategies for coping with stress and fostering resilience, according to the proposals of research questions 2 and 3, respectively.

A new framework of relationships between companies/brands and consumer has arisen. While the isolation period has tested the psychological and mental health of people, brands have developed a new social utility. Therefore, they have innovated and made extra creative efforts, as companies have had to discover the balance between the survival of their business and their social engagement in this crisis; in fact, they have even collaborated for crisis communication management in an alternative way.

In summary, this research introduced an innovative approach in the field of health crisis communication. The findings of this research work show that this pandemic has given companies the opportunity to develop new roles and to contribute to society by spreading messages promoting resilience and de-stressing, thus attaining a dual goal: To exercise a social function and to maintain their position in the minds of consumers. Therefore, the main insight of this study is the ability of brands to support public mental and psychological health, even in a traumatic event such the COVID-19 pandemic, and particularly during isolation.

## Figures and Tables

**Figure 1 ijerph-17-08876-f001:**
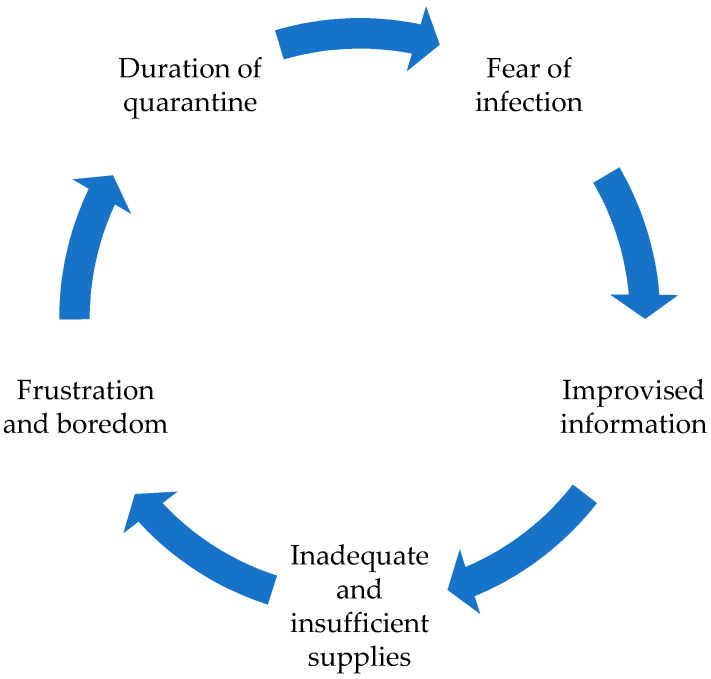
Stress factors during isolation. Source: Own elaboration from Brooks et al., [6].

**Figure 2 ijerph-17-08876-f002:**
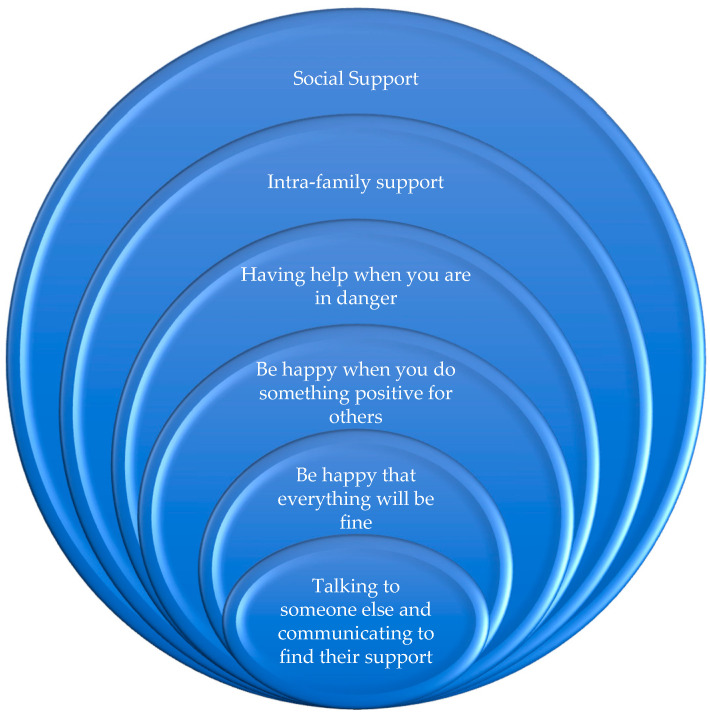
Sources of resilience. Source: Own elaboration from Grotberg [37], Walsh [38], and Adger [39].

**Figure 3 ijerph-17-08876-f003:**
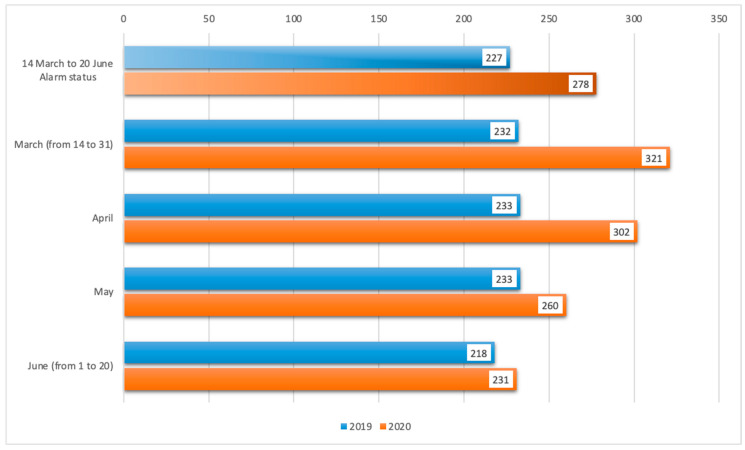
TV consumption in Spain during the alarm state (minutes of person/day) [50].

**Figure 4 ijerph-17-08876-f004:**
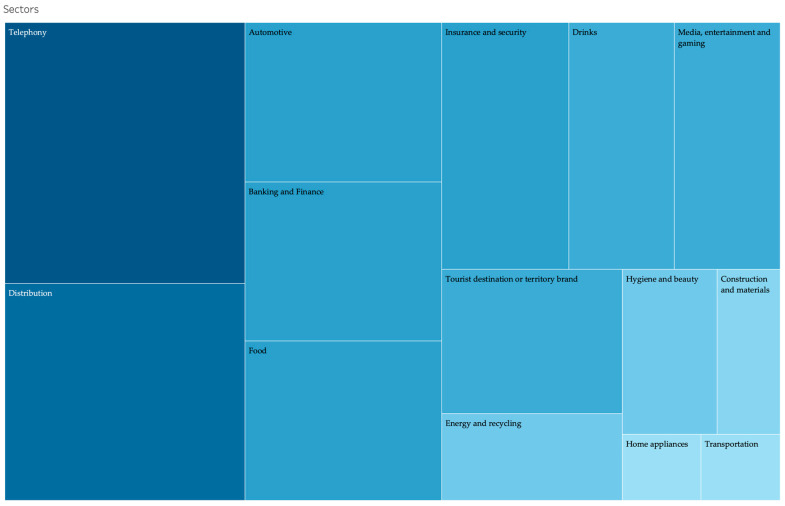
Advertising grouped by sectors.

**Figure 5 ijerph-17-08876-f005:**
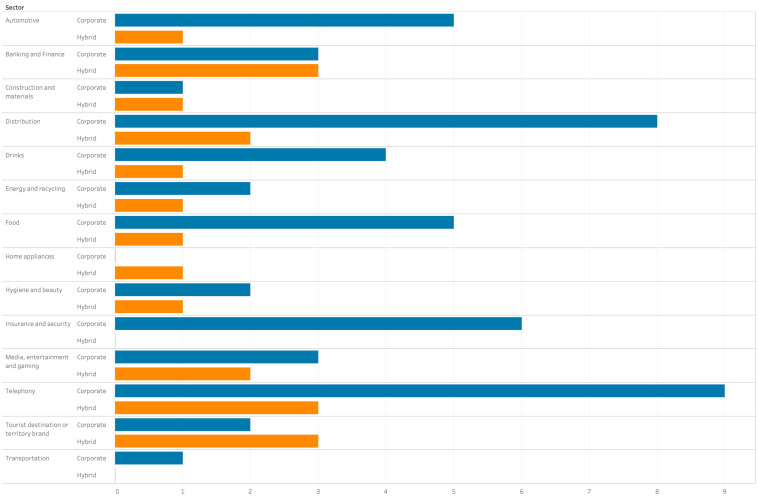
Typology of messages by sector.

**Figure 6 ijerph-17-08876-f006:**
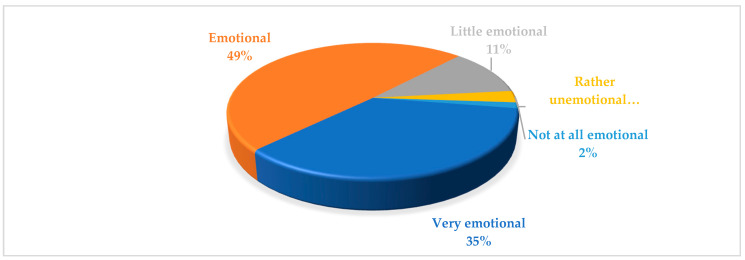
Burden charge of the messages.

**Figure 7 ijerph-17-08876-f007:**
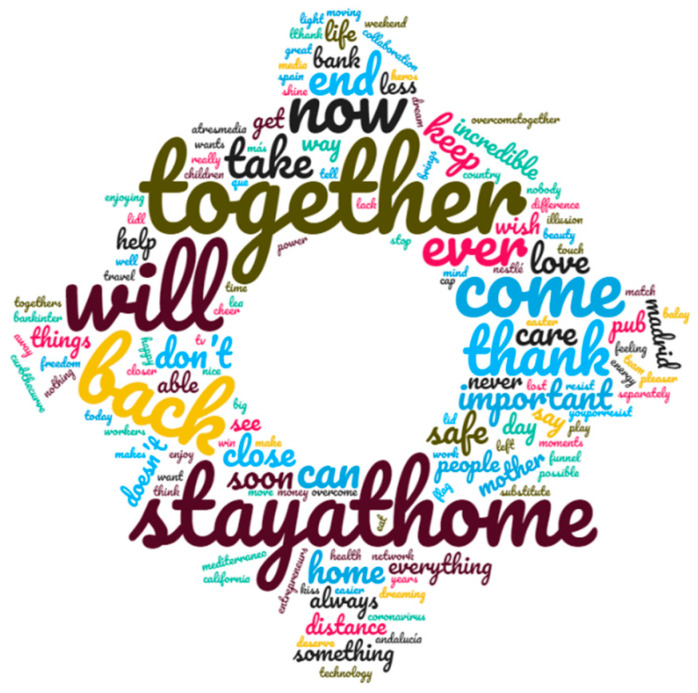
Word cloud of slogans.

**Figure 8 ijerph-17-08876-f008:**
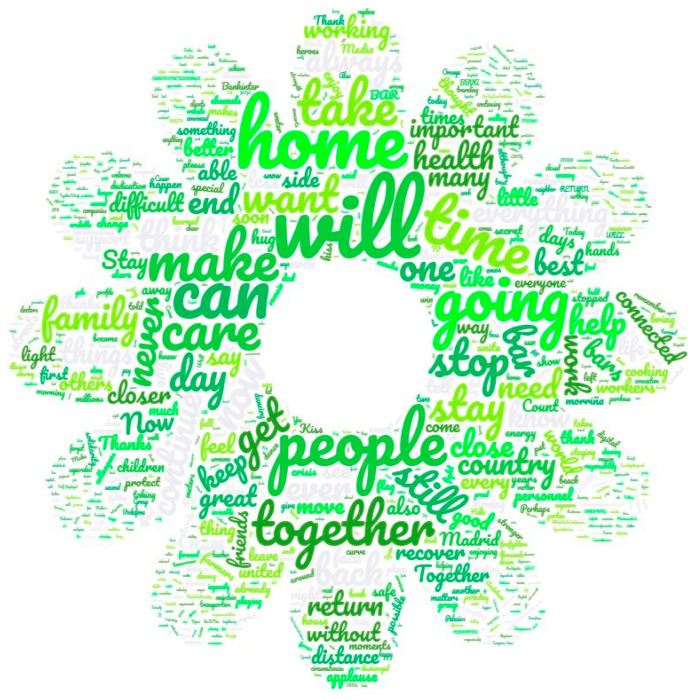
Word cloud of full texts of advertisements.

**Figure 9 ijerph-17-08876-f009:**
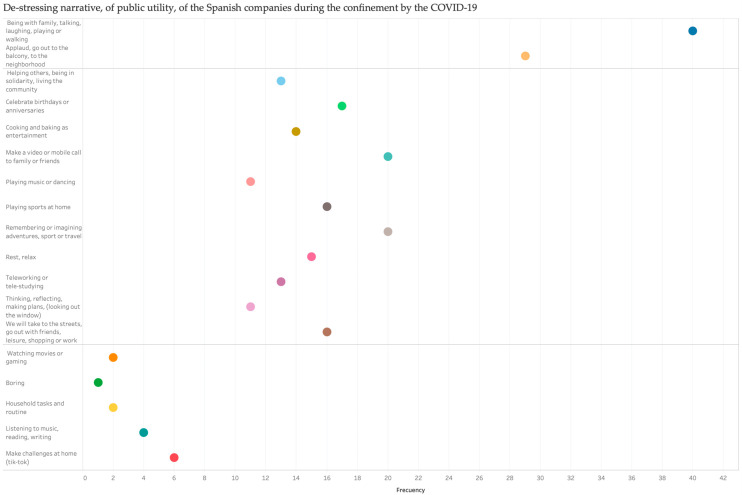
Group of communicative resources with de-stressing components.

**Figure 10 ijerph-17-08876-f010:**
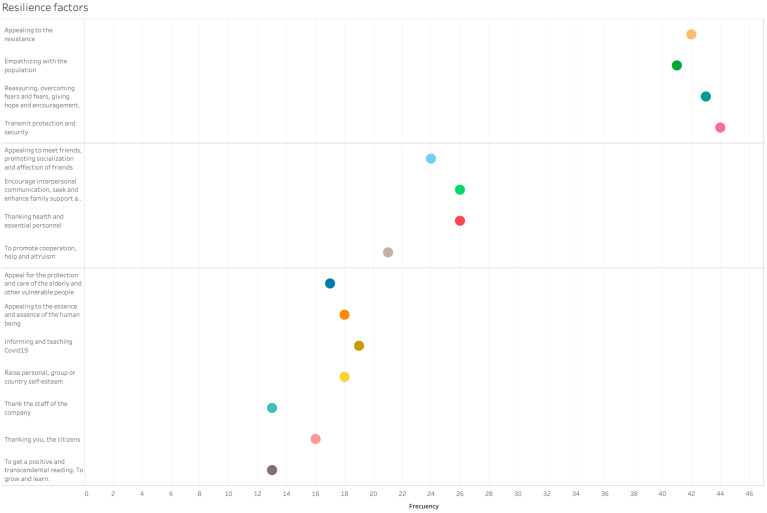
Groups of resilience factors in advertising spots.

**Table 1 ijerph-17-08876-t001:** Social function of advertising in Spain. Source: Own elaboration, Barometer COVID-19 [56,57].

The Expected Role of Brands through Their Advertising	Percentage of the Spanish Population That Agrees with This Information
Advertising should show how companies can be useful in the new daily life	83%
Advertising should inform your efforts to confront the situation	81%
Advertising should use a reassuring tone	79%
Do not take advantage of the coronavirus to promote the brand	75%
Providing a positive perspective	70%
Communicating brand values	62%

**Table 2 ijerph-17-08876-t002:** Analysis content sheet.

N°	Variable	Categories and Values
2 MED	Communication media	Generalist free-to-air televisions (TV1, La 2, Antena 3, La Sexta, Telecinco, Cuatro)
3 SEC	Duration of the advertisement	Seconds
5 BRA	Brand	Product brand name
6 SET	Sector of activity	1. Food
		2. Drinks
		3. Distribution
		4. Energy and recycling
		5. Automotive
		6. Telephony
		7. Hygiene and beauty
		8. Banking and finance
		9. Insurance and security
		10. Tourist destination or territory brand
		11. Media, entertainment, and gaming
		12. Home appliances
		13. Construction and materials
		14. Transportation
7 TIP	Type of message	1. Corporate and institutional or image
		2. Product, service, or sales
		3. Hybrid
8 SLO	Advertisement slogan	The slogan or claim is registered
9 TXT	Full text of the advertisement	The full text is recorded
10 TON	Tone	Very positive
		Positive
		Realistic
		Negative
		Very negative
11 EMO	Emotional burden	Very emotional
		Emotional
		Little emotional
		Rather emotional
		Not at all emotional
12 OFF	Voice over	Text overprinting
		Woman
		Man
		Girl or boy
		Elderly person
13 EST	Stress factors. Factors and coping strategies	1. Rest, relax
		2. Get bored
		3. Thinking, reflecting, planning, looking out the window
		4. Watching movies or gaming
		5. Listening to music, reading, writing
		6. Playing music or dancing
		7. Cooking and baking as entertainment
		8. Being with family, talking, laughing, playing, or walking
		9. We will take to the streets, go out with friends, leisure, shopping, or work
		10. Household tasks and routine
		11. Teleworking or tele-studying
		12. Clap your hands, go out to the balcony, to the neighbourhood
		13. Helping others, being in solidarity, living the community
		14. Make a video or mobile call to family or friends
		15. Playing sports at home
		16. Remembering or imagining adventures, sport, or travel
		17. Doing challenges at home (e.g., using tik-tok)
		18. Celebrate birthdays or anniversaries
14 RES	Resilience: Sources of Resilience	1. Appealing to the resistance
		2. Reassuring, overcoming fears and fears, giving hope and encouragement.
		3. Transmit protection and safety
		4. Thank the health and essential personnel
		5. Thanking the company’s staff
		6. Thanking you, the citizens
		7. Informing and doing pedagogy regarding COVID-19
		8. Raise personal, group, or country self-esteem
		9. Get a positive and transcendental reading. To grow and learn.
		10. To promote co-operation, help, and altruism
		11. Encourage interpersonal communication, seek and enhance family support and affection
		12. Appealing to meet friends, promoting socialization and affection of friends
		13. Empathizing with the population
		14. Appeal to the essence and essence of the human being
		15. Appeal for the protection and care of the elderly and other vulnerable people

**Table 3 ijerph-17-08876-t003:** Messages to overcome stress factors in audiovisual discursive narratives during isolation. Source: Own elaboration from Brooks et al., 2020 [6].

Stress Factors during Confinement	De-Stressing Narratives of Public Utility from Spanish Companies and Brands during the Confinement Due to COVID-19
Duration of quarantine	“It will all be over soon”. “Our scientists are working to find a solution soon”. “Soon it will be back to normal”. “With occupations and distractions, time goes by sooner”.
Fear of infection	“At home you are safe”. “Don’t go out if you don’t have to”. “Be very careful”. “Follow the rules and recommendations”.
Inadequate and insufficient supplies	“We work to guarantee the supply of essential goods to Spanish society”. “We work day and night so that you don’t lack anything”. “We are at your service.
Improvised information	“We keep you informed”. “Use masks, gels and social distance”.

**Table 4 ijerph-17-08876-t004:** Sources of resilience in brand advertising messages. Source: Own elaboration from Connor [35], Walsh [38], Cyrulnik [36], and Grotberg [37].

	Sources of Resilience	Resilient Narratives of Public Utility from Spanish Companies and Brands during the Confinement Due to COVID-19
1	Appealing to the resistance	“Resist. Stay. Hold on. It is what it touches. Take charge”
2	Reassuring, overcoming fears and fears, giving hope and encouragement.	“Don’t worry”. “You are not alone”. “Normality will soon return”. “There is less left”. “Everything will be fine. “Everything will be like before”. “There will be time to do everything”. “We will go back to doing the life of before”
3	Transmit protection and safety	“We are here”. “We work for you”. “Together (we and you) and United”. “You are Safe”
4	Thank the health and essential personnel	“Come out to the balcony and applaud our health heroes, policemen and essential professions”
5	Thanking the company’s staff	“Thanks to you, our workers”. “Our workers are also heroes and essential”
6	Thanking you, the citizens	“Thank you for staying home”. “You are a hero too. “You have our admiration”
7	Informing and doing pedagogy regarding COVID-19	“Stay at home”. “Physical distance, hand washing and hydroalcoholic gel”. “Follow the recommendations of the authorities”
8	Raise personal, group, or country self-esteem	“You are amazing”. “You are extraordinary”. “We are unique”. “We are a great country”.
9	Get a positive and transcendental reading. To grow and learn.	“We realized that we had to stop”. “We were not on the right track”. “There is no evil that does not come from good”. “Something has to change”. “Reflect. Grow up. Think. Change”. “Value more what you have”
10	To promote co-operation, help, and altruism	“Helping makes you feel good”. “Help your neighbours, those who are alone and those who need it most”. “Think about the others”. “Think of the other”.
11	Encourage interpersonal communication, seek and enhance family support and affection	“Talk to your family”. “Lean on others”. “Create bonds with your children, partner and parents”. “Do not distance yourself emotionally from them: make a video call, remember or imagine a re-encounter”
12	Appealing to meet friends, promoting socialization and affection of friends	“Talk to your friends.” “Lean on them”. “Keep the bonds of affection.” “Don’t distance yourself emotionally from them: video calls with friends”
13	Empathizing with the population	“We understand you”. “We are with you: we are part of the” “We “. “We know what you are going through”. “We, the companies, take care of it”. “It doesn’t have to be easy for you”. “We row in the same direction”. “We feel identified. “Your problems are our problems”
14	Appeal to the essence and essence of the human being	“Life. Health. Care. Protection. Affection”. “The small things”. “The small gestures”. “The simple life”. “The simplicity”. “The small pleasures”. “Human being”. “To be united, to create and to maintain the emotional bonds”
15	Appeal for the protection and care of the elderly and other vulnerable people	“Our elderly, people with disabilities and children need you: they are the most vulnerable”

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
