# Peer review of "Resilience and Anti-Stress during COVID-19 Isolation in Spain: An Analysis through Audiovisual Spots"

_ijerph, 2020, doi:10.3390/ijerph17238876_

Round 1

Reviewer 1 Report

The article undertakes a very current and interesting topic. The article is quite organised. But I have some remarks and questions:

  1. Is the paper pioneering in this area of research? I can not find any dates.
  2. Why the Authors decided about the choice of 71 spots? What kind of  criteria?
  3. How to answer the last question -   RQ3. Have companies and their brands incorporated messages that could serve as coping strategies against the psychosocial stress produced by the pandemic?
  4. The article is not sufficiently embeded in the literature.
  5. What does it mean mixed methods (abstract and page number 6)?
  6. Introduction has to be improved.
  7. Practical implications  - very good idea, but it has to b improved.

Author Response

Answers to REVIEWER 1

The article undertakes a very current and interesting topic. The article is quite organised. But I have some remarks and questions:

  1. Is the paper pioneering in this area of research? I can not find any dates.

Dates and specific work have been included to show better that the paper is pioneering.

  1. Why the Authors decided about the choice of 71 spots? What kind of  criteria?

The 71 spots are all the commercials broadcast during the isolation period (lockdown). The researchers chose the whole “population” of ads to provide a more holistic view of visual and textual content related to this unprecedented moment for both society and brands.

We have introduced clarification about it in the sample subsection in Methods.

  1. How to answer the last question -   RQ3. Have companies and their brands incorporated messages that could serve as coping strategies against the psychosocial stress produced by the pandemic?

We agree that we do not pay sufficient attention to answer RQ3. We had supposed that it would have been understood but the current version includes a specific sentence to support and explain the results, discussion and conclusions related to RQ3. Furthermore, we have commented directly on the results in connection with all RQs to provide the contribution of the study clearly.

  1. The article is not sufficiently embedded in the literature.

The manuscript has been improved with  a large amount of literature to better frame the literature.

  1. What does it mean mixed methods (abstract and page number 6)?

We have combined qualitative and quantitative methods to provide an holistic vision of the field of study. As has been explained in a paragraph of the Method section, the content analysis is recognized as qualitative method but some of the variables are defined with a Likert scale in order to develop more complex quantitative analysis as correlation and multivariate analysis with a cluster.

  1. Introduction has to be improved.

The review of the Introduction has been made. The improvement is specially derived from the proof editing and adding some key issues to provide a better framework of the study.  

  1. Practical implications  - very good idea, but it has to be improved.

We have added some extra comments to complete the practical implications of the study. We hope it was in line with that which reviewer 1 proposed.

Reviewer 2 Report

This project is highly timely and interesting, and well-researched. However, as written, the manuscript poorly communicates the research that was conducted and its findings.

Major recommendation: Significant English language support needed to ensure that readers understand the study and findings. The reviewer found it difficult to read and adequately review this article.

The reviewer has attempted to provide sample edits that will help strengthen the article. The entire manuscript should be revised to ensure that readers understand the authors' intent. 

Author Response

Firstly, thank you for your interest in our research work. You have brought up some good points and we appreciate the opportunity to clearly explain our study and its contribution better. Moreover, A proof editing of the English has been done according to your advice.

Round 2

Reviewer 1 Report

Thank you for the all clarification and improvment of the paper. I would like to accept in the present form.

Reviewer 2 Report

Thanks you for the revision! The article is significantly improved. The reviewer has noted some additional edits that would improve the final manuscript. Please see comments in the attached PDF. The reviewer does not need to review again.

Another recommendation is to increase the size of the fonts in the figures and graphs to be more readable/legible.
